# An Improved Harris Hawks Optimization Algorithm and Its Application in Grid Map Path Planning

**DOI:** 10.3390/biomimetics8050428

**Published:** 2023-09-15

**Authors:** Lin Huang, Qiang Fu, Nan Tong

**Affiliations:** 1Faculty of Electrical Engineering and Computer Science, Ningbo University, Ningbo 315211, China; 2111082144@nbu.edu.cn; 2College of Science and Technology, Ningbo University, Ningbo 315300, China; tongnan@nbu.edu.cn

**Keywords:** grid map, path planning, Harris Hawks Optimization algorithm, circle map, random guidance strategy, improved sine-trend search, nonlinear jump strength

## Abstract

Aimed at the problems of the Harris Hawks Optimization (HHO) algorithm, including the non-origin symmetric interval update position out-of-bounds rate, low search efficiency, slow convergence speed, and low precision, an Improved Harris Hawks Optimization (IHHO) algorithm is proposed. In this algorithm, a circle map was added to replace the pseudo-random initial population, and the population boundary number was reduced to improve the efficiency of the location update. By introducing a random-oriented strategy, the information exchange between populations was increased and the out-of-bounds position update was reduced. At the same time, the improved sine-trend search strategy was introduced to improve the search performance and reduce the out-of-bound rate. Then, a nonlinear jump strength combining escape energy and jump strength was proposed to improve the convergence accuracy of the algorithm. Finally, the simulation experiment was carried out on the test function and the path planning application of a 2D grid map. The results show that the Improved Harris Hawks Optimization algorithm is more competitive in solving accuracy, convergence speed, and non-origin symmetric interval search efficiency, and verifies the feasibility and effectiveness of the Improved Harris Hawks Optimization in the path planning of a grid map.

## 1. Introduction

Intelligent agent path planning is currently one of the hottest research topics. In fact, path planning has been widely used in multiple fields, such as mobile robots [1,2,3], UAVs [4], and urban planning [5]. Especially in smart factories, efficient path planning can significantly save time and reduce costs. Path planning can be considered a combinatorial optimization problem. The solution to this type of problem is usually based on traditional algorithms and intelligent algorithms. Traditional algorithms include the A* (A-star) algorithm [6,7], Dijkstra algorithm [8], artificial potential field [9], Rapidly-exploring Random Trees (RRT) [10,11], etc., whereas intelligent algorithms include the Particle Swarm Optimization (PSO) [12], Genetic Algorithm (GA) [13], Grey Wolf Optimizer algorithm (GWO) [14], Artificial Fish Swarm Algorithm (AFSA) [15], Whale Optimization Algorithm (WOA) [16], etc. Due to the simple structure, fast convergence speed, high search precision, and simple operation of the swarm intelligence algorithm, it is currently used in many practical projects [17,18]. Wang et al. proposed an adaptive and balanced Grey Wolf Optimization algorithm, which uses a novel level-based strategy to select random wolves to improve the exploration energy of the algorithm and introduces an adaptive coefficient to dynamically adjust the exploration and exploitation capabilities of different optimization stages. It is used to find the optimal feature subset for high-dimensional classification [19]. Liu et al. proposed a differential evolutionary chaotic Whale Optimization Algorithm, which increases the diversity of the population by introducing the initial population of sine chaos theory. A new adaptive inertia weight is introduced to improve the optimization performance of the algorithm, and the differential variance algorithm is integrated to improve the search speed and accuracy of the algorithm. And, the improved algorithm is applied to the distribution network fault location of the IEEE-33 nodes [20]. Tajziehchi et al. proposed a new approach for selecting optimal accelerographs, and the scaling of them for dynamic time history analysis is presented by the binary genetic algorithm and natural numbers in order to achieve the mean response spectrum, which has a proper matching and a short distance with the target spectrum and indicates the expected earthquake of the site [21]. In the literature, there has been a lot of research on 2D path planning and mobile robots in recent years [1,2,3], especially research on 2D mobile robot path planning [22,23,24,25,26]. The mainstream of mobile robot path planning is based on the grid map. The classification of 2D path planning methods is generally divided into classical algorithms and natural heuristic algorithms. Since classical algorithms cause problems such as high computational cost and low computational efficiency, heuristic algorithms have a fast response speed and are more suitable for the optimization of local path planning [27]. Therefore, the natural heuristic algorithm is the most suitable method for two-dimensional grid map path planning.

The Harris Hawks Optimization algorithm is a swarm intelligence algorithm proposed by Heidari et al. in 2019 [28]. This algorithm simulates the special hunting behavior of the Harris hawk population. The HHO algorithm has a total of six phases: two exploration phases and four exploitation phases. In order to find the optimal solution, HHO randomly executes one of the six stages. Due to its powerful structure, its original performance is far superior to many recognized methods, such as the PSO, GWO, AFSA, and WOA algorithms [29]. Compared with these popular methods, the results of HHO are of relatively high quality. This feature also supports why it is widely used [30]. At the same time, the HHO algorithm also has the characteristics of good convergence speed, powerful neighborhood search, good balance between exploration and exploitation, suitability for many problems, easy implementation, and so on [31]. Therefore, HHO has been applied to convolutional neural network structures [32], image segmentation [33,34], UAV path planning [35], and PM prediction [36], among other fields. However, similar to other swarm intelligence algorithms, the Harris hawks algorithm also has its own drawbacks, such as being prone to falling into local optima, jumping out of the interval in non-origin symmetric intervals, and having low convergence accuracy. With the emergence of more and more highly complex problems, many scholars have tried to improve HHO to enhance its performance. Zhang et al. explored six escaping energy and proposed MHHO, which proves the superiority and effectiveness of the exponential decreasing strategy [37]. Liu et al. introduced square neighborhood topology and random arrays to solve the imbalance problem in the HHO exploration phase and exploitation phase and improve the algorithm’s convergence accuracy and optimization ability [38]. Zhu et al. proposed a method based on chemotaxis correction to improve the HHO algorithm, improving the problem of slow convergence speed and the tendency to fall into local optimum [39]. Sihwail et al. proposed a new search mechanism using an elite backward learning strategy to avoid falling into local optima and enhance search ability [40]. Zhang et al. proposed the ADHHO algorithm based on adaptive coordinated foraging and dispersed foraging to improve the stability of the algorithm because the HHO algorithm has low population diversity and a single search method during the exploration phase [41]. Although these improvements made to the HHO algorithm have achieved good results, it is worth noting that there is no perfect algorithm capable of handling all the optimization problems due to the No Free Lunch (NFL) theory [42]. Moreover, these improvements are not suitable for handling nonsymmetric interval path planning problems.

As engineering problems become more complex and automation becomes more widespread, the environment that path planning technology faces will become more complex too, requiring it to adapt to the environment. However, using traditional strategies, such as the A* (A-star) algorithm and the Dijkstra algorithm, to implement path planning is far from sufficient. Currently, many scholars have conducted research on path planning by applying swarm intelligence algorithms [43] and other strategies. For example, Liu X et al. designed a hybrid path planning algorithm based on optimized reinforcement learning and improved particle swarm optimization. This algorithm optimizes the hyper-parameters of reinforcement learning to make it converge quickly and learn more efficiently. A preset operation was designed for the particle swarm algorithm to reduce the calculation of invalid particles, and the algorithm proposes a correction variable obtained from the cumulative reward of reinforcement learning; this corrects the adaptation of the individual optimal particle and the global optimal position of the particle swarm optimization degree to obtain effective path planning results. Also designed was a selection parameter system to help choose the optimal path [22]. Lab X et al. proposed a method based on the improved A-star ant colony algorithm, using the A-star algorithm to optimize the initial pheromone of the ant colony to improve the convergence speed of the algorithm and, at the same time, increase the enhancement coefficient of the ant colony pheromone to avoid the late stage of the ant colony algorithm’s excess accumulation of pheromones, leading to local optimization problems [23]. Szczepanski R et al. used the artificial bee colony algorithm to optimize the path globally and then used the Dijkstra algorithm to select the shortest path, making it possible to obtain at least a satisfactory path in the workspace with a small computational workload in real-time path planning [24]. Xiang D et al. improved the evaluation function of the A* algorithm, increased the convergence speed of the A* algorithm, and removed the unnecessary nodes in the A* algorithm, thereby reducing the path length and used the mechanism of the greedy algorithm to insert unselected nodes one by one. In the optimal path, the planning of multiple target points is realized at one time [25]. Zhang Z et al. proposed an improved sparrow search algorithm, which added a new domain search strategy and a new position update function to improve the fitness value and convergence speed of the global individual, respectively, and introduced it to obtain high-quality paths and fast convergence, and proposed a linear Path strategy, so that the robot can reach the goal faster [26]. Although the aforementioned fusion research has achieved good results in the path planning of mobile robots, there are also some shortcomings. For example, path planning is implemented by rounding the coordinates when following the grid map, which may sacrifice the length of the path. And, using random points for path planning can not only search for paths in a wider range but also improve the flexibility of path planning and avoid local optimal solutions. The way of randomly picking points can improve globality, flexibility, and robustness.

The current path planning interval is based on map data, and the maps usually use asymmetric origin intervals to represent data distribution. Path planning maps provide an optimal path based on the asymmetric origin interval between the starting point and the endpoint and are asymmetrical. The origin interval will have a certain impact on the algorithm of the path planning map. Due to the asymmetric origin interval between the starting point and the endpoint, the path lengths in different regions are different. Therefore, the algorithm of the path planning map needs to take this into account to ensure that the final path is the shortest one and goes through a suitable area. In the standard HHO algorithm, when facing non-origin symmetric intervals, the position updating formula during the exploration phase may cause the HHO algorithm to jump out of the defined interval, thereby reducing the search efficiency. Because MHHO is biased toward the exploration phase, it improves the formula of the exploration position, making MHHO more likely to jump out of the set range. ADHHO proposed a phase based on adaptive cooperative foraging, dispersed foraging, and a biased exploitation phase, which will make the jump out of the set interval much smaller. However, the position step size improvement is not complete, resulting in a high probability of jumping out of the set interval. Based on the above problems, this paper improves the HHO algorithm; using the circle map during the population initialization reduces the probability of initializing points on the boundary and prevents out-of-bounds during position updating, thus increasing the purpose of the initial population and improving search efficiency. The use of nonlinear jump strength improves the convergence accuracy of the local search stage. Integrating the concept of the A* algorithm into the fitness function of the Harris Hawks Optimization algorithm to plan the path of the 2D grid map, the grid map is no longer limited to the selection and rounding of the center of the grid, but to the random selection of points, so that the path length is shorter and more stable.

In the experimental section, the out-of-bounds rate of the IHHO algorithm is compared with HHO and its variants in asymmetric intervals. A total of 11 benchmark functions are selected to perform the function optimization tasks, compare each strategy of the IHHO separately, and compare with other algorithms. Through the statistics and analysis of the experimental data, the Improved Harris Hawks Optimization algorithm has better performance in solution accuracy, convergence speed, and non-origin symmetric interval search efficiency.

The rest of this paper is organized as follows: In Section 2, a mathematical optimization model for path planning in the grid map is given. In Section 3, the basic principle of HHO is introduced. Section 4 introduces the IHHO algorithm in detail. Section 5 simulates and experiments on the IHHO algorithm and verifies the performance of the IHHO algorithm. Section 6 applies the IHHO algorithm to the path planning of the grid map and verifies its superiority and effectiveness. Section 7 concludes with some conclusions and some prospects for the future.

## 2. Description of the Mathematical Problem

To study the path planning problem of the grid map, it is necessary to establish a grid map model that is similar to the environment. The obstacles are classified as black blocks, while white blocks represent safe areas. When determining the optimization objective function (fitness function), the path length is the major consideration of the optimization objective.

### 2.1. Path Length

The path length is the primary consideration in path planning problems, as it affects the robot’s running time and energy consumption. The premise of seeking the shortest path is to avoid obstacles. The calculation formula for the path length is shown in Formula (1), where $x_{i}$ is the horizontal coordinate of the node and $y_{i}$ is the vertical coordinate of the node.
(1)$$f\left(x\right)=\sum_{i=1}^{N-1}\sqrt{{\left(x_{i}-x_{i+1}\right)}^{2}-{\left(y_{i}-y_{i+1}\right)}^{2}}$$

### 2.2. Mathematical Model

The A* algorithm is a commonly used algorithm for path planning. The A* algorithm mainly calculates the goodness of the traversed nodes using the evaluation function [44]. The core expression of the evaluation function is shown in Formula (2).
(2)F(n)=G(n)+H(n)

The evaluation function is set to *F*, and the value of F(n) can estimate the minimum path cost from the initial node to any node *n* and the minimum path cost from node *n* to a destination node. G(n) is defined as knowing the actual cost from the start node to *n* nodes, and H(n) is defined as the actual cost of the optimal path from *n* nodes to the target node [45].

For an optimal path cost f(x) from the initial node to the target node, it will be divided into two parts, namely g(x) and h(x); g(x) is the moving cost from the initial node vector to the target node vector, that is, the path length, and h(x) is the path cost of the initial node vector and the target node vector. Then, the mathematical optimization model of the path length cost function f(x) can be defined as follows:(3)minf(x)=ming(x)+minh(x).

By combining the path length and cost function into the optimization objective function, it is obvious that the desired better path in the grid map path planning process is one that is both the shortest and crosses fewer obstacles.

### 2.3. Grid Modeling

The principle of the grid method is to divide the entire two-dimensional space into grids of the same size and simulate the obstacles and safety areas as a collection of small squares of different colors, which form a grid map. Figure 1 is the simulated 2D modeling of the path planning environment. The position of the obstacle is marked as the black part, the white area is the feasible area, “o” represents the starting point, and “x” represents the endpoint.

## 3. Standard Harris Hawks Optimization (HHO) Algorithm

The HHO algorithm simulates the behavior of Harris hawks while hunting rabbits. The algorithm attacks prey through strategies such as assaulting, encirclement, and disturbance. HHO mainly includes two phases, exploration and exploitation, and the selection between the two phases is based on the absolute value of the prey’s escape energy *E*. Figure 2 shows the different phases.

### 3.1. Exploration Phase

In the Harris Hawks Optimization algorithm, individual Harris hawks are considered candidate solutions, and the best candidate solution at each iteration is treated as the target prey or a near-optimal solution. During the exploration phase, Harris hawks search for prey in two different ways, which are defined as follows:(4)$$X\left(t+1\right)=\left\{\begin{array}{cc} X_{rand}\left(t\right)-r_{1}\left|X_{rand}\left(t\right)-2r_{2}X\left(t\right)\right| & \;\mathrm{if}\;q\geq 0.5\\ \left(X_{rabbit}-X_{m}\left(t\right)\right)-r_{3}\left(LB+r_{4}\left(UB-LB\right)\right) & \;\mathrm{if}\;q<0.5 \end{array}\right.,$$
where $X_{rand}\left(t\right)$ denotes a random selection of the individual Harris hawk’s position and $X_{rabbit}\left(t\right)$ is the prey location or optimal location for the Harris hawk population. $r_{1}$, $r_{2}$, $r_{3}$, $r_{4}$, and *q* are all random numbers within the interval (0,1), where $r_{1}$, $r_{2}$, $r_{3}$, and $r_{4}$ are scaling coefficients used to provide diverse trends and *q* is used to simulate the random selection of prey. UB and LB represent the upper and lower bounds of the search space, respectively. $X_{m}\left(t\right)$ denotes the average position of the Harris hawk population, which is calculated using the following formula:(5)$$X_{m}\left(t\right)=\frac{1}{N}\sum_{i=1}^{N}X_{i}\left(t\right).$$

### 3.2. Exploitation Phase

During the exploitation phase, in order to trap prey, the Harris hawk employs four different exploitation strategies based on the prey’s behavior. Using a random number *r* within the range (0,1) to assume if the prey has successfully escaped, if r<0.5, it means that the prey successfully escapes the encircling circle; otherwise, it means that the prey fails to escape the encircling circle. When $\left|E\right|<0.5$, the hawk will use the hard besiege method, while, when $0.5\leq \left|E\right|<1$, it will use the soft besiege method. When $0.5\leq \left|E\right|<1$ and r≥0.5, meaning that the prey has enough energy to successfully escape, the Harris hawk updates its position using the soft besiege strategy. The specific calculation formula is as follows:(6)$$X\left(t+1\right)=△X\left(t\right)-E|JX_{rabbit}\left(t\right)-X\left(t\right)|$$
(7)$$△X\left(t\right)=X_{rabbit}\left(t\right)-X\left(t\right)$$
(8)$$J=2(1-r_{5}),$$
where △X(t) represents the difference in the position vectors between the prey and the current position vector, *J* is the random jump strength during the prey’s escape process, and $r_{5}$ is a random number within the range (0,1).

When $0\leq \left|E\right|<0.5$ and r≥0.5, which means the prey has insufficient energy but can still successfully escape, the Harris hawk uses a hard besiege method for the position update. The specific calculation formula is as follows:(9)$$X\left(t+1\right)=x_{rabbit}-E\left|△X(t)\right|.$$

When $0.5\leq \left|E\right|<1$ and r<0.5, which means the prey has sufficient energy but failed to escape the encirclement, the Harris hawk uses a soft besiege with the progressive rapid dives method for the position update. The specific calculation formula is as follows:(10)$$X\left(t+1\right)=\left\{\begin{array}{cc} Y=X_{rabbit}-E\left|JX_{rabbit}-X\left(t\right)\right| & \;\mathrm{if}\;F(Y)<F(X(t))\\ Z=X_{rabbit}-E\left|JX_{rabbit}-X\left(t\right)\right|+S\times LF\left(D\right) & \;\mathrm{if}\;F(Z)<F(X(t)) \end{array}\right.,$$
where *D* is the problem dimension, *S* is a 1×D random variable, and *F* is the fitness function. LF is the levy flight function. The specific calculation formula is as follows:(11)$$LF\left(x\right)=0.01\times \frac{\mu \sigma }{{\left|\nu \right|}^{\frac{1}{\beta }}},\sigma ={\left(\frac{\Gamma \left(1+\beta \right)\times \sin\left(\frac{\pi \beta }{2}\right)}{\Gamma (\frac{1+\beta }{2}\times \beta \times 2^{\frac{\beta -1}{2}})}\right)}^{\frac{1}{\beta }},$$
where μ and ν are random numbers within the range (0,1) and β is a default constant of 1.5.

When $0\leq \left|E\right|<0.5$ and r<0.5, this means the prey has low escape energy and the hawk continually approaches the prey to reduce the distance. If the prey does not have enough energy to escape, the hawk will try to shorten the central position between itself and the fleeing prey. At this time, the Harris hawk will use a hard besiege with the progressive rapid dives method for the position update. The specific calculation formula is as follows: (12)$$X\left(t+1\right)=\left\{\begin{array}{cc} Y=X_{rabbit}\left(t\right)-E\left|JX_{rabbit}-X_{m}\left(t\right)\right| & \;\mathrm{if}\;F(Y)<F(X(t))\\ Z=X_{rabbit}\left(t\right)-E\left|JX_{rabbit}-X_{m}\left(t\right)\right|+S\times LF\left(D\right) & \;\mathrm{if}\;F(Z)<F(X(t)) \end{array}\right.,$$
where $X_{m}\left(t\right)$ is calculated using Formula (5) and LF is computed using Formula (11).

### 3.3. Escaping Energy

The exploratory and development stages can be converted through the escaped energy *E* of the prey, and the absolute value of the escaped energy *E* decreases continuously during the process of evading the pursuit of the Harris hawk. The calculation formula of the escaped energy *E* is shown below:(13)$$E=2E_{0}\left(1-\frac{t}{T}\right),$$
where $E_{0}$ represents the initial state of the prey, which is a random number between (−1,1); *t* is the current iteration number; and *T* is the maximum iteration times. The variation of the escaped energy with the iteration times is shown in Figure 3.

## 4. Improved Harris Hawks Optimization (IHHO) Algorithm

The standard HHO is a stable and high-performance optimization algorithm that can be used to solve many practical engineering problems. Firstly, in the global search phase, only random individuals are used for position updates, and there is a lack of information exchange between populations, resulting in a susceptibility to premature convergence and an inability to escape local optima. Secondly, HHO relies on the average position of the population, which leads to a reduction in diversity in the early phase and may ignore the existence of the optimal solution. Thirdly, the position update formula for the exploration stage has a large step size, which makes it easy to go out-of-bounds during the search phase, especially in non-origin-symmetric intervals. Fourthly, the jump strength of the prey mainly affects the development stage and the random jump strength has a certain probability of causing too large local search step sizes and increasing the instability of the algorithm. By using the nonlinear convergence curve combined with the escaped energy to control the step size, the randomness of the algorithm can be better controlled. In addition, the jump strength of the prey should be proportional to its escaped energy. The lower the energy, the lower the height or distance the prey can jump. Therefore, this paper adopts the following strategies:1.Using the circle map to initialize the population to reduce the number of initialization points on the boundary.2.Introducing the random guidance strategy and improvement sine-trend search strategy to replace Formula (Equation 4), which increases the information exchange between populations, improves diversity, and reduces the step size and premature convergence to some extent, thus increasing the efficiency of the global search of the hawks. The improved sine-trend search strategy reduces the dependence of the population on the average position and guides the individual hawks to approach the prey, thus improving the convergence in the exploration stage.3.Proposing the nonlinear jump strength convergence strategy, which combines the random jump strength with the prey’s escape energy to increase the convergence accuracy during a local search.

### 4.1. Circle Map

Both the chaotic map and pseudo-random strategy possess randomness and unpredictability, but the chaotic map has a certain regularity and is often used instead of the pseudo-random strategy for better results. Many chaotic maps are available, such as the circle map, Chebyshev map, intermittency map, iterative map, logistic map, sine map, sinusoidal map, tent map, and singer map. In order to place the initial position of the Harris hawk as close to the center as possible, we use the circle map to initialize the population. Its mathematical model is as follows:(14)$$X_{K+1}=mod\left(X_{K}+a-\frac{b}{2\pi }\sin\left(2\pi X_{K}\right),1\right),$$
where mod() is a mod function, $X_{1}$ is obtained from a random function with the interval (0,1), *K* is an integer, a=0.5, and b=0.2.

### 4.2. Random Guidance Strategy

In Formula (Equation 4) of the exploration phase, using the same random position for updating the current position leads to a lack of communication between the populations and greatly increases the out-of-bounds rate during a search. By calculating the vector differences between multiple random positions and the current position, on the one hand, the out-of-bounds rate can be reduced and, on the other hand, the information exchange between the populations can be increased. This avoids a single global optimal position and prevents the early convergence of HHO to some extent. At the same time, it helps the algorithm to further explore and improves the efficiency of a global search. The formula for calculating the random guidance factor is as follows:(15)$${▽}_{i}=X_{i}\left(t\right)-X\left(t\right),$$
where $X_{i}\left(t\right)$ is the vector position of the random Harris hawk. The vector difference is used as the step size to guide the Harris hawk population to update their positions. The specific formula for the position update is as follows:(16)$$X\left(t+1\right)=X\left(t\right)+\frac{\eta }{n}\sum_{i=1}^{n}{▽}_{i},$$
where η is the vector weight and *n* is the number of random Harris hawks.

### 4.3. Improved Sine-Trend Search

In the exploration phase of the standard HHO, the position updates are based on the average position of the population and the updating of the positions in the exploitation phase also depends on the average position of the population. Therefore, when the population is trapped in a local optimum, it hardly produces a new position. When the HHO faces non-origin-symmetric intervals, the position update is easily out of the defined interval, making the position update invalid and reducing the search efficiency.

The Sine Cosine Algorithm (SCA) [46] uses adaptive amplitude changes in sine and cosine functions for global and local search, so that individual Harris hawks can search around themselves, reducing the out-of-bound rate and increasing optimization performance.

As the Harris hawks algorithm is currently in the exploration phase, only the exploration phase of the sine cosine strategy is retained and the exploitation phase of the sine cosine strategy is not used. The formula for the improved sine search is shown as follows:(17)$$X\left(t+1\right)=X\left(t\right)+\omega \sin\left(\phi \right)\left|r_{6}X_{rabbit}-X\left(t\right)\right|,$$
where ω is a nonlinear convergence coefficient, ϕ is a phase factor, and $r_{6}$ is a random number between 0 and 1. The formulas for ω and ϕ are as follows:(18)$$\omega =2(1-\frac{t}{T})$$
(19)$$\phi =r_{7}+\frac{\pi }{6}+P\pi ,$$
where $r_{7}$ is a random number between 0 and 1 and *P* is an integer of 0 or 1. In order to increase the convergence of the algorithm and guide individuals closer to the population’s optimal value, the golden ratio coefficient [47] is introduced into the formula to search around the vicinity of the optimal value. The formula for the improved sine-trend search strategy is shown as follows:(20)$$X\left(t+1\right)=X\left(t\right)+\omega \sin\left(\phi \right)\left|cX_{rabbit}-dX\left(t\right)\right|,$$
where *c* and *d* are coefficients obtained by incorporating the golden ratio, c=−1+2(1−τ), d=−1+2τ, and the golden ratio numbers $\tau =\frac{\sqrt{5}-1}{2}$.

### 4.4. Nonlinear Jump Strength

In the HHO algorithm, the escape energy *E* determines the transformation between global search and local search and affects the exploitation phase. The escape energy *E* of the prey shows a linear downward trend during the process of being chased, while the jump strength *J* is a random number of (0,2). It is difficult to describe the actual changing trend of biological energy. Some scholars have found that the law of energy consumption can be introduced into the jump strength of prey by combining the escape energy and random jump strength [48]. The overall trend of the physical energy consumption of the prey during the pursuit is declining. As the prey’s physical strength decreases, the jump strength also gradually decreases, which is more in line with biological laws. And, because the jump strength is the influencing factor of the step size in the exploitation phase, it affects the exploitation phase. The jump strength decreases nonlinearly, thus improving the convergence accuracy of the local search stage. The specific calculation formula for nonlinear jump strength is as follows:(21)$$J=2E^{2}.$$

### 4.5. Computational Complexity

#### 4.5.1. Time Complexity Analysis

The time complexity is an important index to evaluate the operation efficiency of an algorithm. The original paper on HHO [28] has already analyzed the time complexity of HHO and will not discuss these details again. The computation time complexity of HHO mainly lies in three aspects: population initialization, fitness evaluation, and position updates. During population initialization, the HHO generates N×D numbers through pseudo-random generation, while IHHO generates N×D numbers through the circle map. The time complexity of both methods for initializing the population is O(N). When updating the position vectors, HHO has a minimum time complexity of O(N×T) and a maximum time complexity of $O\left(N\times T\right)+O\left(N\times \frac{T}{2}\right)$. On the other hand, IHHO does not add a position update formula, so the time complexity for both methods for updating positions is the same.

#### 4.5.2. Performing Step Time Complexity Analysis

The steps to analyze complexity are listed as the following:Step 1: During the initial population calculation, the N×D numbers need to be computed, with a computation complexity of O(N).Step 2: The fitness value of the individuals in the population needs to be evaluated once, with a computation complexity of O(1).Step 3: The escape energy needs to be calculated once, with a computation complexity of O(1).Step 4: If the progressive encirclement approach is used, the position needs to be updated twice; otherwise, it only needs to be updated once, resulting in a computation complexity of O(1+step). If the progressive encirclement approach is used then step=1; otherwise, step=0.Step 5: The fitness value and the overall optimal value need to be updated twice, with a computation complexity of O(2).The overall execution time complexity is O(N+(5+step)×T).

According to the improvement idea mentioned above, the algorithm flow chart of the IHHO is shown in Figure 4. The pseudo code of the IHHO is as follows Algorithm 1:
**Algorithm 1** Pseudo-code of IHHO algorithm.**Input:** The population size and maximum**Output:** The location of rabbit and its fitness value
1:Initialize the random population2:**while** 
t<T 
**do**3:     Calculate the fitness values of hawks4:     Set $X_{rabbit}$ as the location of rabbit (best location)5:     **for** each hawk ($X_{i}$) **do**6:           Update the initial energy $E_{0}$ and jump strength *J*7:           Update *E* using Equation (Equation 13)8:           **if** $\left|E\geq 1\right|$ **then**9:              Update the location vector using Equation (Equation 16) or Equation (Equation 17)10:          **else**11:              **if** $0.5\leq \left|E\right|<1$ and r≥0.5 **then**12:                  Update the location vector using Equation (Equation 6)13:              **else if** $0.5\leq \left|E\right|<1$ and r<0.5 **then**14:                  Update the location vector using Equation (Equation 10)15:              **else if** $0\leq \left|E\right|<0.5$ and r≥0.5 **then**16:                  Update the location vector using Equation (Equation 9)17:              **else**18:                  Update the location vector using Equation (Equation 12)19:              **end if**20:           **end if**21:       **end for**22:**end while**23:**return** 
$X_{rabbit}$


## 5. Algorithm Simulation Experiment and Analysis

All experiments were implemented in Matlab 2018b. All computations were run with the following hardware: Intel Core i7-8550u, 1.80 GHz, 8 GB RAM, and Windows 10 (64-bit) operating system. The algorithm parameter settings used in the simulation experiment are shown in Table 1.

### 5.1. Out-of-Bounds Comparison

The grid map is a typical non-origin symmetric path planning map. The standard HHO algorithm sets the search step too large in non-origin-symmetric intervals during the exploration phase, which leads to easy position updates that venture outside of the set interval, resulting in ineffective position updates and reduced algorithm efficiency. In contrast, IHHO balances global search and controls the step size during the exploration phase, reducing the probability of venturing outside the set interval and improving algorithm efficiency. This section sets the maximum number of iterations *T* to 500, the dimensionality *D* to 30, the population *N* to 30, and the interval to [0, 100], and conducts 20 function tests, with the final mean taken as the result. As shown in Figure 5, the number of out-of-bounds occurrences is shown in Table 2.

From Figure 5, it can be seen that in the early phase of iteration, the number of out-of-bounds occurrences in HHO continues to increase and, after transitioning from the exploration phase to the development phase and entering local search, the number of out-of-bounds occurrences no longer increases. However, IHHO consistently maintains a low number of out-of-bounds occurrences throughout the exploration phase. A clear comparison can be made through Table 2, which shows that using the improved algorithm can significantly reduce the Harris hawk positions venturing outside the set interval and improve the efficiency of the Harris hawk position updates.

It can be seen from Figure 5 that the inflection point of the line segment is the transformation between the exploration phase and the exploitation phase. The IHHO, HHO, and their variant algorithms are analyzed from the exploration and exploitation behaviors. Exploration and exploitation are two important aspects of the algorithm. When the number of iterations is fixed, if the algorithm is biased toward exploration, the algorithm will find the extreme points of the function faster, but the accuracy will be lost by the algorithm, and it will be easier to jump out of the set range. If the algorithm is biased toward exploitation, the number of out-of-bounds will be reduced, but the algorithm is prone to fall into the local optimum. Therefore, the balance between exploration and exploitation needs to be considered. In general, the results are better when the number of iterations of the two phases is close [49]. As in the introduction and analysis of this paper, IHHO is improved based on non-origin symmetric interval problems. It is a good way to favor the exploitation phase, but the HHO algorithm itself has the disadvantage of being easily trapped in the local optimum. Based on the above, and the analysis of the experimental data, IHHO performance is at its best when the ratio between the exploration and exploitation is close to unity.

### 5.2. Test Function

In this experiment, 11 benchmark functions with different optimization characteristics were randomly selected from the commonly used CEC test functions, including five unimodal benchmark functions, three multimodal benchmark functions, and three fixed-dimensional benchmark multimodal functions. The IHHO algorithm was tested and compared using these benchmark functions, where F1–F5 are unimodal benchmark functions and F6–F11 are multimodal benchmark functions, all of which can be used to test the optimization performance and convergence accuracy of the algorithm. The test functions are listed in Table 3.

In order to better verify the impact of each strategy on the IHHO algorithm, this section uses 11 benchmark functions to compare the circle map, improved sine-trend search, random guidance strategy, nonlinear jump strength, and IHHO. In order to be able to make a fair comparison, the number of populations *N* is set to 30 and the maximum number of iterations *T* is 500. Each group was individually tested for 30 rounds and its average value, optimal value, worst value, and standard deviation were recorded. The test results are shown in Table 4.

From Table 4, and from the test results of the F1–F4 unimodal functions, the improved sine-trend search has the greatest impact on IHHO. According to the test results of function F5, the circle map, improved sine-trend search strategy, and nonlinear jump strength have a greater impact on IHHO than the random guidance strategy. The test results of function F6 show that the impact of nonlinear jump intensity on IHHO is the largest. The impact on functions F7–F8 is similar. According to the test results of functions F9 and F11, the random guidance strategy and the improved sine-trend search strategy have the largest impact on IHHO, while function F10 is the one with the largest impact on IHHO by random guidance strategy and nonlinear jump strength.

From the overall analysis of Table 4, the IHHO algorithm performance of all strategies fusion is optimal, followed by the improved sine-trend search strategy, but the improved sine-trend search strategy will lead to an easier fall into the local optimum because of its tendency to optimality, while the random guidance strategy increases the mutual communication of its population, which can better improve this shortcoming. From the test and results, the circle map and nonlinear jump strength provide support to IHHO in terms of optimization accuracy.

In this section, HHO, GWO [14], WOA [16], PWOA [50], MHHO [37], ADHHO [41], and IHHO were compared using the 11 benchmark functions. To ensure a fair comparison, the population size *N* was set to 30, and the maximum number of iterations *T* was set to 500 for each function test. Each function was tested separately for 30 rounds, recording its mean, best, worst, and standard deviation results; the test results are shown in Table 5.

From Table 5, it can be seen that the IHHO algorithm in this paper achieved 11 optimal results on the benchmark test function. The results demonstrate that the improved HHO algorithm is superior to the other five algorithms. On the unimodal test function, the convergence speed of IHHO is the fastest among the other six algorithms, further verifying the effectiveness of the proposed improvements in improving the convergence speed of the HHO algorithm. On the multimodal test function, in the case of test function F6, it can be seen from the standard deviation that IHHO is the most stable among the compared algorithms, including HHO, MHHO, and ADHHO. As for test functions F7 and F8, IHHO performs equally well as HHO, MHHO, and ADHHO. In terms of the fixed-dimensional multimodal benchmark functions F9 and F10, the numerical differences between the test functions are not significant. However, it can still be clearly observed that IHHO has higher stability compared to the other algorithms. Furthermore, based on test function F11, IHHO has improved significantly compared to the other improved and original HHO, while GWO has higher stability on this test function. Overall, the IHHO algorithm maintains a fast search speed and has certain advantages in stability for multimodal functions. On the test functions, it is obvious that IHHO has superior convergence performance and stability compared to the other algorithms. The fitness convergence curves on some test functions are shown in Figure 6 and Figure 7.

In Figure 6, the convergence curve and functions F1 to F4 show that IHHO has higher accuracy and can converge to the global optimal solution more quickly than the other five algorithms. Function F5 converges the fastest in the early stage but falls into a local optimum in the middle stage and jumps out of it in the later stage, obtaining better values than the other five algorithms. Function F6 converges to the optimal value first and maintains a better value than the other five algorithms throughout.

In Figure 7, functions F7 and F8 show that IHHO has excellent search performance and can converge to the optimal value faster than the other five algorithms. Function F9 has the highest initial function value due to the effect of the chaotic map, falls into a local optimum in the middle stage, but eventually jumps out of it, making IHHO the algorithm with the best optimal value ranking among the six algorithms. Functions F10 and F11 show that IHHO performs better than the other comparative algorithms in terms of optimization performance.

Based on the comprehensive analysis of Table 5 and Figure 6 and Figure 7, IHHO has good optimization performance, the ability to jump out of local optima, and higher convergence accuracy. Based on the number of optimal values, IHHO ranks first with 11 optimal values among the six algorithms. In terms of comprehensive convergence accuracy, and optimization performance among the six algorithms, IHHO still ranks first.

## 6. Grid Map Path Planning

The interval of path planning is a typical non-origin symmetric interval, and the standard HHO algorithm has insufficient search efficiency for non-origin symmetric intervals. However, IHHO has made certain improvements for non-origin symmetry. By combining the HHO’s own position updating mechanism, the IHHO has better advantages and higher stability in path planning on a grid map.

### 6.1. Fusion of A* and IHHO Algorithm

The A algorithm is a commonly used algorithm for path planning. A mainly calculates the quality of the traversal nodes through an evaluation function, which is shown in Equation (Equation 2), where G(n) represents the path length as a movement cost and H(n) represents the cost function, whose formula is as follows:(22)$$G\left(N\right)=\sum_{i=1}^{N-1}\sqrt{{\left(x_{i}-x_{i+1}\right)}^{2}+{\left(y_{i}-y_{i+1}\right)}^{2}}$$
(23)H(N)=ck,
where *N* represents the population size, *c* represents the number of obstacles crossed, and *k* is a constant value, which we set as the area of the path planning map. The path length and cost function are combined as the evaluation function (fitness function). In the process of path planning on a grid map, a large constant value is added to the path length for each obstacle crossed as the cost function, and the fitness function is the sum of the path length and cost function, as shown in Formula (24).
(24)$$F\left(N\right)=\sum_{i=1}^{N-1}\sqrt{{\left(x_{i}-x_{i+1}\right)}^{2}+{\left(y_{i}-y_{i+1}\right)}^{2}}+ck$$

### 6.2. Parameter Setting of Grid Map Path Planning

To better highlight the superiority of the improved algorithm, this paper introduces the EGWO [51], HIWOA [52], HHO, MHHO, ADHHO, and IHHO algorithms for path planning. The introduced EGWO and HIWOA have similar partial test results on the benchmark function. For fairness, the other algorithms adopt the same strategies as IHHO path planning, such as A-star and elite retention. This paper uses two different grid maps, 20×20 and 60×60, so, for the 20×20 grid map, the number of populations (nodes) is set to 20, according to the horizontal (vertical) of the grid, the maximum iteration times of the path is set to 50, and the maximum iteration times within the population is set to 500. Whereas for the 60 × 60 grid map, the number of populations is set to 60, according to the horizontal (vertical) coordinates of the grid. Due to more information, the maximum iteration times of the population is increased to 1000. The parameter settings in the introduced EGWO and HIWOA algorithms are based on Table 1. The specific parameter settings are shown in Table 6.

### 6.3. Experimental Results and Analysis of Path Planning

Figure 1 shows the models of the environments. The starting points are both set at (0.5, 0.5) and the endpoints are set at (19.5, 19.5) and (59.5, 59.5), respectively. The black grids represent the obstacles and the rest are safe zones. The results of the improved algorithm and the six introduced algorithms on the 20×20 grid map path planning are shown in Figure 8 and Figure 9. The results of the path planning on the 60×60 grid map are shown in Figure 10. The 20×20 path comparison convergence graph is shown in Figure 11a. and the path comparison convergence graph is shown in Figure 11b. For each experiment, it was conducted 10 times independently and the average value, optimal value, and worst value were recorded. The results are shown in Table 7.

From Table 7, it can be seen that A-star has deficiencies in path planning for random point selection and complex scenarios, but adding swarm intelligence algorithms to A-star can effectively solve this problem.

In Figure 8 and Figure 9, the IHHO algorithm has the shortest path length among all the algorithms in a 20×20 grid. Compared to the other algorithms, IHHO has achieved good results not only in path selection, but also in path length optimization. In Figure 10, the IHHO algorithm has the best path optimization in a 60×60 grid, while other algorithms perform poorly in path length optimization, and the path length increases with an increasing number of grids and obstacles. Based on the comparison in Figure 8, Figure 9 and Figure 10, as well as Table 7, it can be concluded that the IHHO algorithm has the shortest path length and the most stable path. IHHO has the best optimization effect, followed by HHO, EGWO, and ADHHO, while HIWOA and MHHO have the worst optimization effect.

In Figure 11a, the IHHO and EGWO algorithms showed excellent optimization performance in the early stages of searching for paths, followed by the ADHHO and HHO algorithms. Throughout the later stages, the IHHO algorithm maintains the shortest path length, while the A-star algorithm only finds the better path later. In Figure 11b, the IHHO and HHO algorithms demonstrated good path optimization performance, followed by the EGWO and MHHO algorithms. In the later stages, the IHHO algorithm showed signs of a continuous optimization of the path length.

The EGWO algorithm borrows the modified position update function from Particle Swarm Optimization (PSO) and makes the GWO algorithm be guided in the position updates with the tent map, which gives it an advantage in diversity in the initial population and, thus, makes it better in the early stage of path planning than HIWOA. HIWOA introduces the inertia weight coefficient, which makes the position update smaller and introduces a nonlinear convergence factor, increasing its convergence accuracy. The HIWOA algorithm also introduces a feedback mechanism that makes the algorithm more capable of escaping local optima, making the HIWOA algorithm better than the EGWO algorithm in the later stage of path planning.

The MHHO and ADHHO algorithms adjust the escape energy, which makes the algorithms biased toward searching, and the path searching range is wide but the performance of local convergence is reduced. The ADHHO algorithm introduces adaptive collaboration and dispersion foraging strategies, which increases its search for paths and corrects the escape energy to enhance the performance of local convergence in the exploitation phase. However, overall, compared with the ADHHO algorithm, the IHHO algorithm still has better convergence and search effectiveness.

## 7. Conclusions

This article analyzes the effectiveness of the basic HHO algorithm in finding that there are shortcomings in the search effectiveness of the nonzero point symmetric interval. Meanwhile, it is discovered that the basic HHO algorithm has problems with convergence accuracy and convergence speed. This paper proposes an improved HHO algorithm. First, the circle mapping in chaotic mapping is used to initialize the population, increasing its purposefulness and enabling the distribution of the population to be centered, thus reducing the probability of location updates exceeding the bounds and improving the diversity and effectiveness of the algorithm. Secondly, a random direction strategy and an improved sine tendency search strategy are used to improve the exploration stage, completely replacing the global search formula of the original algorithm and constructing a composite novel search formula, effectively improving the effectiveness and efficiency of the algorithm in searching the nonzero point symmetric interval. Finally, the combination of escape energy and jump strength is used to improve the convergence accuracy of the algorithm.

To verify the superiority and feasibility of the new algorithm, first, on the benchmark test function, the four strategies were compared and tested to test the impact of each strategy on the IHHO algorithm. The experimental results show that the four strategies have good performance on IHHO, especially the improved sine-trend search. Then, we compared and tested the GWO, PWOA, HHO, MHHO, ADHHO, and IHHO algorithms. The experimental results show that the IHHO algorithm is generally superior to the compared algorithms, such as GWO, WOA, MHHO, ADHHO, and the basic HHO algorithm, in terms of convergence accuracy, convergence speed, and stability. Finally, we applied IHHO to the path planning of the grid map. In the experimental results, the optimal length of the path planning length of the IHHO algorithm is 1.61–21.06% lower than that of the EGWO, HIWOA, HHO, MHHO, and ADHHO algorithms in Scenario 1, and the average length of the path planning in Scenario 2 is 5.98–13.03% lower than that of the other algorithms. The length is 4.37–9.75% lower than other algorithms, indicating that the IHHO planning ability is significantly better than the EGWO, HIWOA, MHHO, and ADHHO algorithms. The algorithm still has room for optimization, and the path planning is based on a known and fixed grid map. There are still many challenges for future work, such as the continuous optimization of the algorithm, dynamic obstacles, and unknown interference in path planning. Therefore, the improvement of the HHO algorithm and the problem of grid path planning will continue to receive attention. In addition, it can be seen from the experiment that as the map continues to grow, the optimization begins to be difficult, and the effect of randomly picking points is not obvious compared with the center rounding. It can be concluded that this method is suitable for smaller grid maps. Of course, with the continuous improvement of equipment and facilities, we will continue to optimize methods and strive to apply research results to actual environments.

## Figures and Tables

**Figure 1 biomimetics-08-00428-f001:**
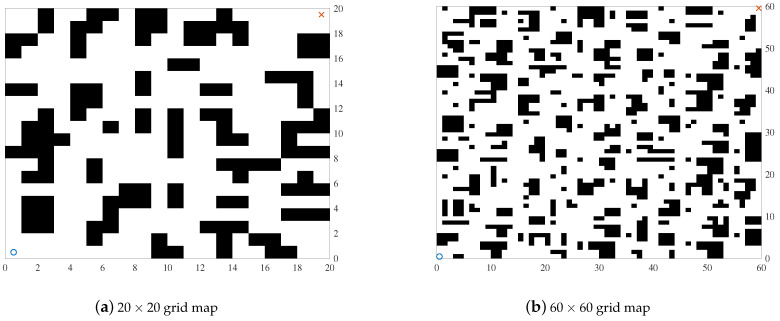
Mathematics model.

**Figure 2 biomimetics-08-00428-f002:**
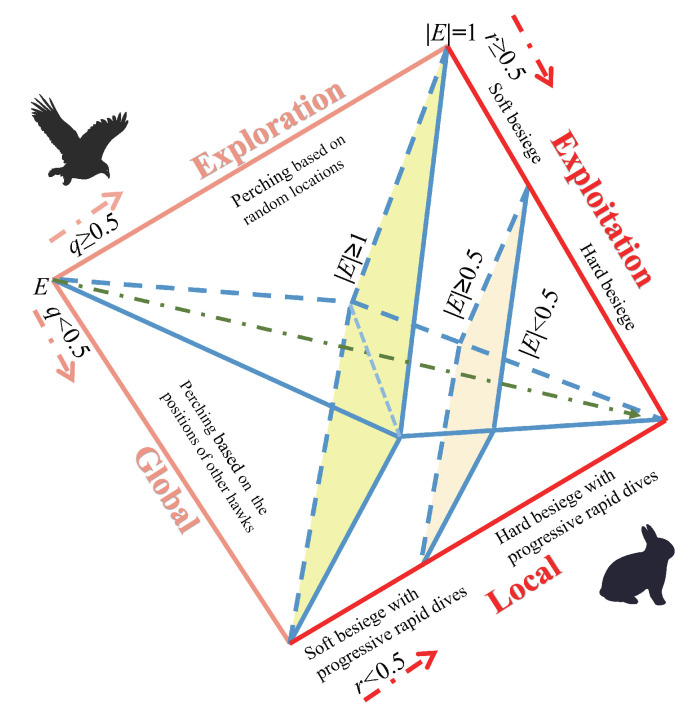
Different phases of HHO.

**Figure 3 biomimetics-08-00428-f003:**
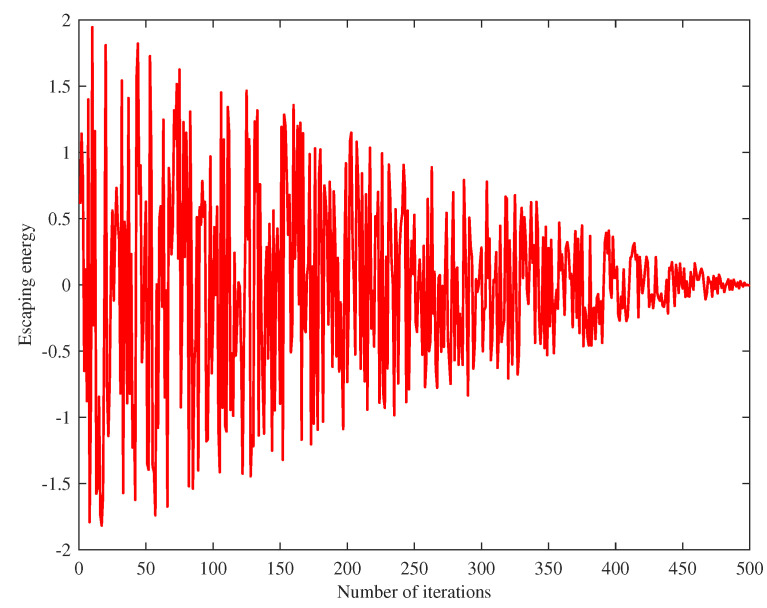
Escaping energy.

**Figure 4 biomimetics-08-00428-f004:**
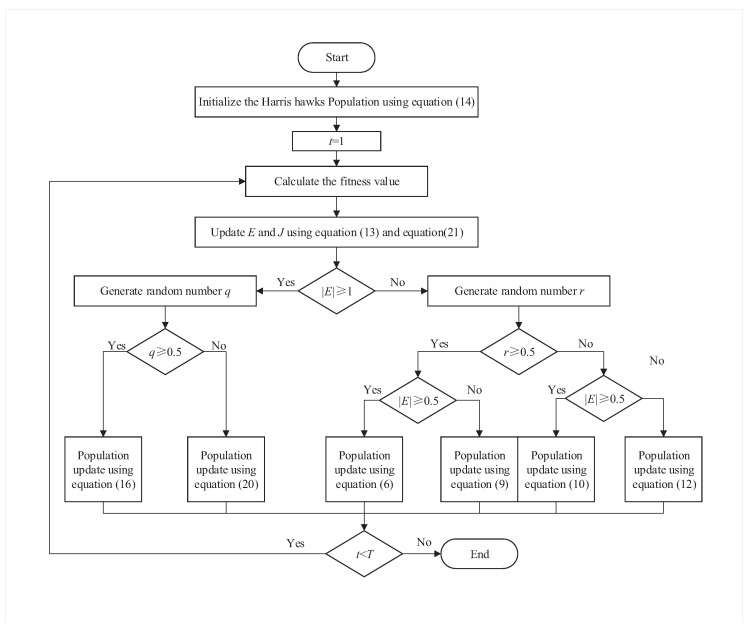
Flow chart of IHHO.

**Figure 5 biomimetics-08-00428-f005:**
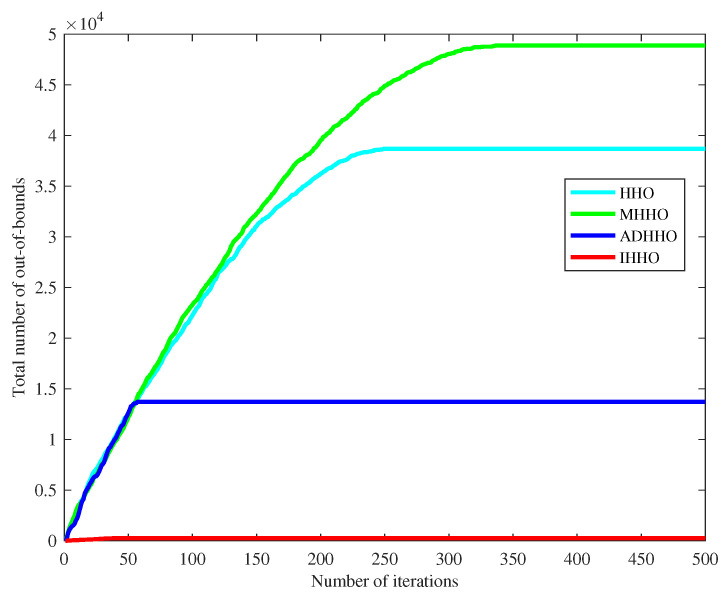
Comparison diagram of out-of-bound numbers.

**Figure 6 biomimetics-08-00428-f006:**
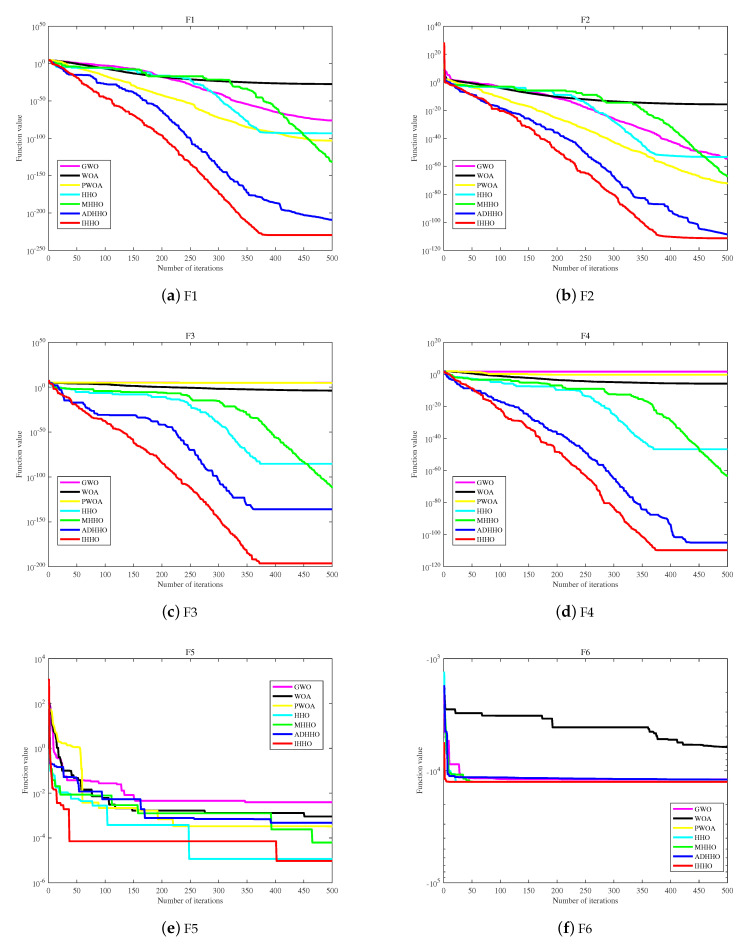
The convergence curves of fitness values of 11 test functions.

**Figure 7 biomimetics-08-00428-f007:**
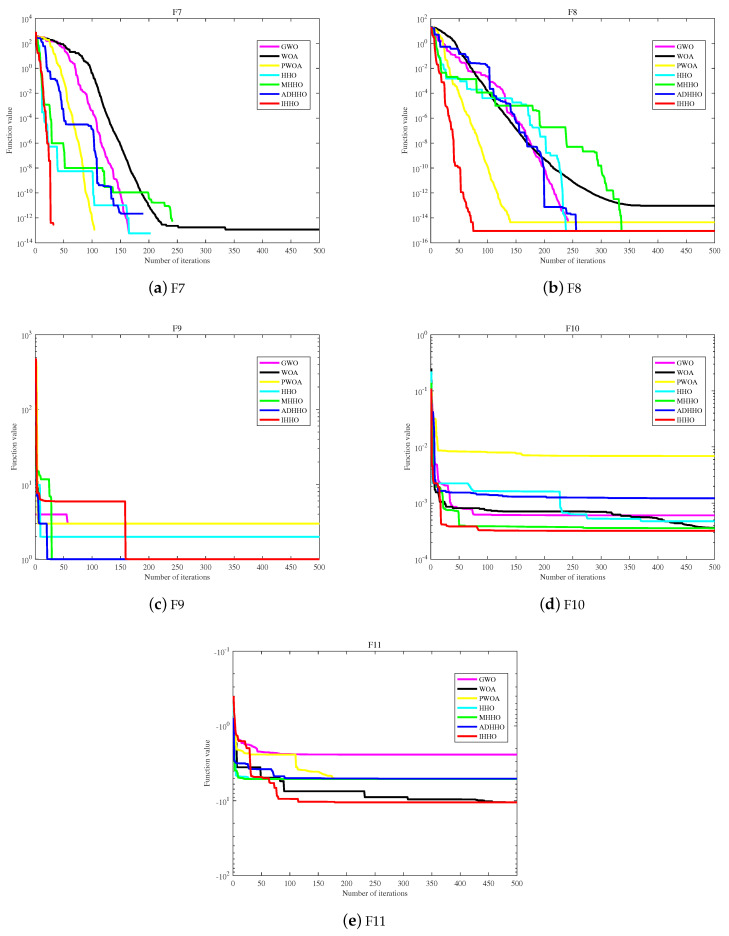
The convergence curves of fitness values of 11 test functions.

**Figure 8 biomimetics-08-00428-f008:**
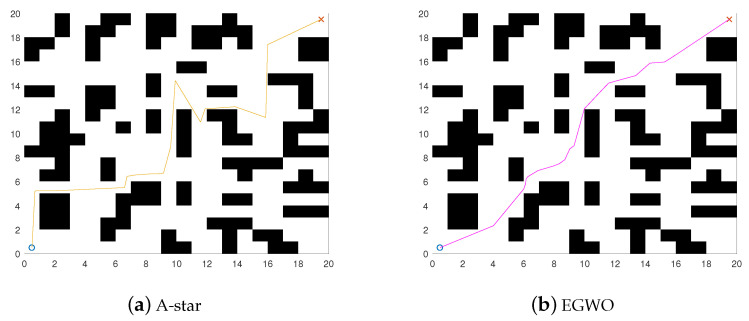
20×20 path comparison diagram.

**Figure 9 biomimetics-08-00428-f009:**
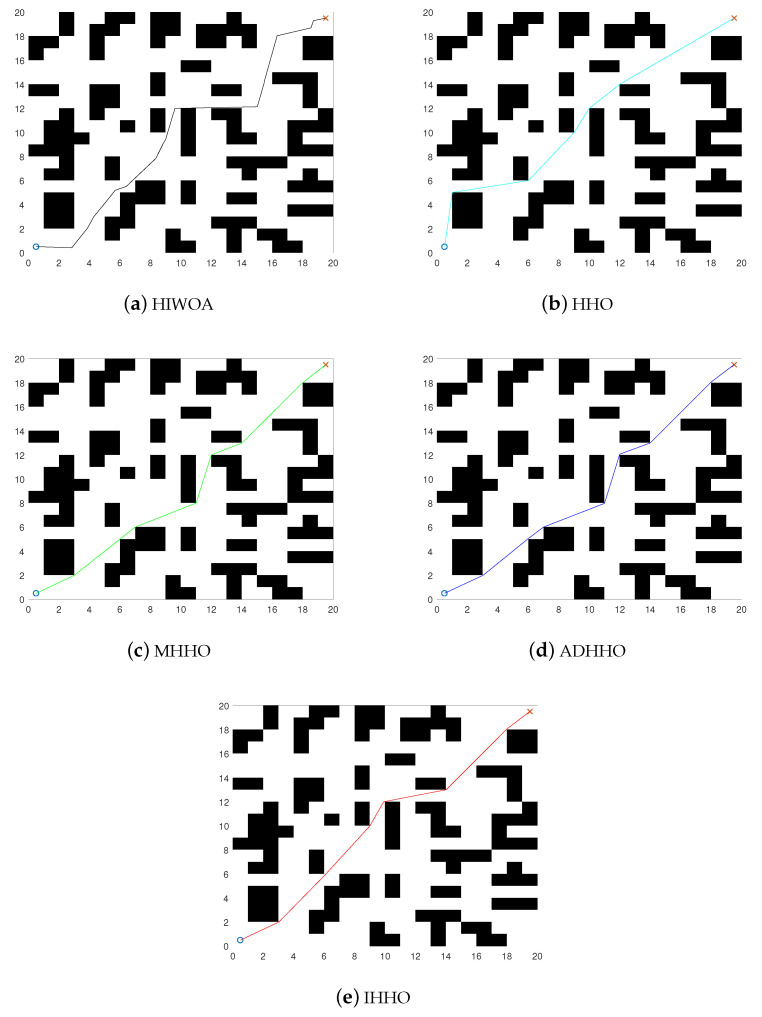
20×20 path comparison diagram.

**Figure 10 biomimetics-08-00428-f010:**
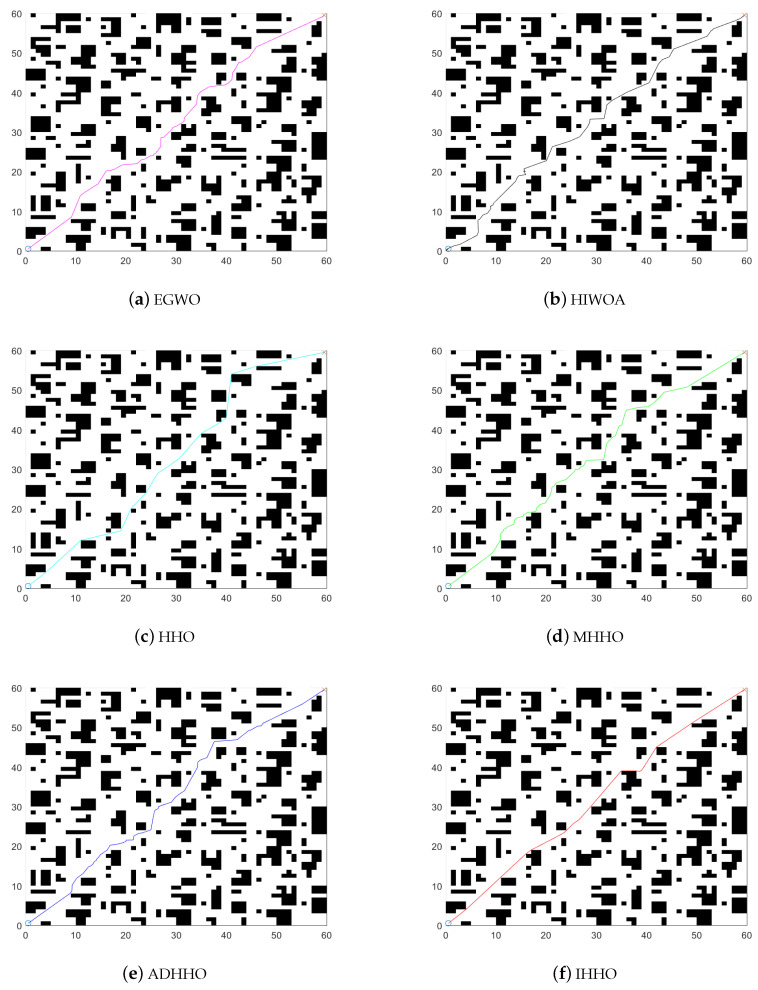
60×60 path comparison diagram.

**Figure 11 biomimetics-08-00428-f011:**
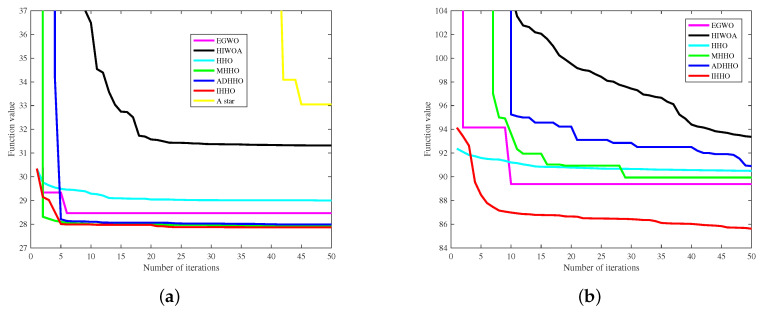
Path comparison convergence graph. (**a**) Seven algorithms compare the convergence graph in the 20×20 path. (**b**) Six algorithms compare convergence graphs in 60×60 path.

**Table 1 biomimetics-08-00428-t001:** Parameter settings of algorithm.

Algorithm	Parameter Name	Parameter Value
PWOA	Learning coefficient α	0.2
	Discount factor γ	0.8
EGWO	Individual memory coefficient b1	0.1
	Communication coefficient b2	0.9
	Control parameter initial value $a_{initial}$ and $a_{final}$	2 and 0
ADHHO	Attenuation factor δ	1.5
IHHO	The number of random Harris hawks *n*	3
	Vector weight η	random(0,1)

**Table 2 biomimetics-08-00428-t002:** Comparison table of total numbers out-of-bounds.

Algorithm	Min	Max	Mean	Out-of-Bound Probability
HHO	36,607	40,777	38,067.00	8.459%
MHHO	44,882	50,476	47,758.50	10.613%
ADHHO	3796	25,201	12,726.75	2.828 %
IHHO	56	638	351.00	0.078%

**Table 3 biomimetics-08-00428-t003:** Benchmark function.

No	Function	Dimension	Interval	$f_{\min}$
F1	$f\left(x\right)=\sum_{i=1}^{n}x_{i}^{2}$	30	[−100,100]	0
F2	$f\left(x\right)=\sum_{i=1}^{n}\left|x_{i}\right|+\prod_{i=1}^{n}\left|x_{i}\right|$	30	[−10,10]	0
F3	$f\left(x\right)=\sum_{i=1}^{n}{\left(\sum_{j=1}^{i}x_{i}\right)}^{2}$	30	[−100,100]	0
F4	$f\left(x\right)=\max\left\{\left|x_{i},1\leq i\leq n\right|\right\}$	30	[−100,100]	0
F5	$f\left(x\right)=\sum_{i=1}^{n}ix_{i}^{4}+random\left[0,1\right)$	30	[−1.28,1.28]	0
F6	$f\left(x\right)=\sum_{i=1}^{n}-x_{i}\sin\left(\sqrt{\left|x_{i}\right|}\right)$	30	[−500,500]	−418.9829 × *n*
F7	$f\left(x\right)=\sum_{i=1}^{n}\left[x_{i}^{2}-10\cos\left(2\pi x_{i}\right)+10\right]$	30	[−5.12,5.12]	0
F8	$f\left(x\right)=-20\exp\left(-0.2\sqrt{\frac{1}{n}\sum_{i=1}^{n}x_{i}^{2}}\right)-\exp\left(\frac{1}{n}\sum_{i=1}^{n}\cos\left(2\pi x_{i}\right)\right)+20+e$	30	[−32,32]	0
F9	$f\left(x\right)={\left(\frac{1}{500}+\sum_{j=1}^{25}\frac{1}{j+\sum_{i=1}^{2}{\left(x_{i}-a_{ij}\right)}^{6}}\right)}^{-1}$	2	[−65.536,65.536]	1
F10	$f\left(x\right)=\sum_{i=1}^{11}{\left[a_{i}-\frac{x_{1}\left(b_{i}^{2}+b_{i}x_{2}\right)}{b_{i}^{2}+b_{i}x_{3}+x_{4}}\right]}^{2}$	4	[−5,5]	0.00030
F11	$f\left(x\right)=\sum_{i=1}^{10}{\left[\left(X-a_{i}\right){\left(X-a_{i}\right)}^{T}+c_{i}\right]}^{-1}$	4	[0,10]	−10.5363

**Table 4 biomimetics-08-00428-t004:** The value of each test function under different strategies.

Function	Index	HHO	Circle Map	Improved Sine-Trend Search	Random Guidance Strategy	Nonlinear Jump Strength	IHHO
F1	mean	1.50 × $10^{-95}$	6.31 $\times \;10^{-99}$	4.64 $\times \;10^{-192}$	3.15 $\times \;10^{-90}$	1.97 $\times \;10^{-99}$	3.25 $\times \;10^{-221}$
	best	1.28 $\times \;10^{-109}$	1.39 $\times \;10^{-117}$	3.38 $\times \;10^{-212}$	2.55 $\times \;10^{-113}$	2.53 $\times \;10^{-123}$	1.62 $\times \;10^{-238}$
	worst	3.63 $\times \;10^{-94}$	1.86 $\times \;10^{-97}$	1.39 $\times \;10^{-190}$	9.44 $\times \;10^{-89}$	5.86 $\times \;10^{-98}$	4.20 $\times \;10^{-220}$
	std	6.75 $\times \;10^{-95}$	3.40 $\times \;10^{-98}$	2.54 $\times \;10^{-191}$	1.72 $\times \;10^{-89}$	1.07 $\times \;10^{-98}$	0.99 $\times \;10^{-220}$
F2	mean	3.95 $\times \;10^{-51}$	8.27 $\times \;10^{-52}$	8.18 $\times \;10^{-91}$	5.38 $\times \;10^{-51}$	6.46 $\times \;10^{-54}$	2.16 $\times \;10^{-108}$
	best	5.45 $\times \;10^{-59}$	8.21 $\times \;10^{-60}$	1.70 $\times \;10^{-106}$	2.85 $\times \;10^{-56}$	1.93 $\times \;10^{-63}$	9.26 $\times \;10^{-124}$
	worst	8.91 $\times \;10^{-50}$	1.50 $\times \;10^{-50}$	2.43 $\times \;10^{-89}$	6.50 $\times \;10^{-50}$	1.80 $\times \;10^{-52}$	6.48 $\times \;10^{-107}$
	std	1.64 $\times \;10^{-50}$	2.99 $\times \;10^{-51}$	4.43 $\times \;10^{-90}$	1.54 $\times \;10^{-50}$	3.28 $\times \;10^{-53}$	1.18 $\times \;10^{-107}$
F3	mean	5.54 $\times \;10^{-71}$	3.33 $\times \;10^{-78}$	6.52 $\times \;10^{-146}$	7.81 $\times \;10^{-68}$	7.56 $\times \;10^{-72}$	3.85 $\times \;10^{-172}$
	best	5.14 $\times \;10^{-103}$	5.40 $\times \;10^{-96}$	2.56 $\times \;10^{-183}$	3.43 $\times \;10^{-97}$	7.08 $\times \;10^{-103}$	3.39 $\times \;10^{-196}$
	worst	1.65 $\times \;10^{-69}$	9.97 $\times \;10^{-77}$	1.74 $\times \;10^{-144}$	2.06 $\times \;10^{-66}$	2.27 $\times \;10^{-70}$	1.15 $\times \;10^{-170}$
	std	3.02 $\times \;10^{-70}$	1.82 $\times \;10^{-77}$	3.18 $\times \;10^{-145}$	3.77 $\times \;10^{-67}$	4.14 $\times \;10^{-71}$	2.10 $\times \;10^{-171}$
F4	mean	3.16 $\times \;10^{-48}$	1.55 $\times \;10^{-50}$	1.51 $\times \;10^{-92}$	2.24 $\times \;10^{-45}$	2.74 $\times \;10^{-53}$	6.19 $\times \;10^{-112}$
	best	1.21 $\times \;10^{-56}$	1.47 $\times \;10^{-59}$	4.59 $\times \;10^{-107}$	5.30 $\times \;10^{-59}$	1.48 $\times \;10^{-61}$	5.35 $\times \;10^{-121}$
	worst	8.18 $\times \;10^{-47}$	2.72 $\times \;10^{-49}$	3.65 $\times \;10^{-91}$	6.70 $\times \;10^{-44}$	4.36 $\times \;10^{-52}$	1.33 $\times \;10^{-110}$
	std	1.49 $\times \;10^{-47}$	5.42 $\times \;10^{-50}$	6.76 $\times \;10^{-92}$	1.22 $\times \;10^{-44}$	9.36 $\times \;10^{-53}$	2.44 $\times \;10^{-111}$
F5	mean	1.58 $\times \;10^{-04}$	1.44 $\times \;10^{-04}$	9.00 $\times \;10^{-05}$	1.46 $\times \;10^{-04}$	1.35 $\times \;10^{-04}$	3.74 $\times \;10^{-05}$
	best	9.70 $\times \;10^{-06}$	4.00 $\times \;10^{-06}$	4.08 $\times \;10^{-06}$	1.41 $\times \;10^{-05}$	6.13 $\times \;10^{-06}$	2.33 $\times \;10^{-06}$
	worst	5.82 $\times \;10^{-04}$	7.22 $\times \;10^{-04}$	2.73 $\times \;10^{-04}$	6.68 $\times \;10^{-04}$	4.15 $\times \;10^{-04}$	9.57 $\times \;10^{-05}$
	std	1.75 $\times \;10^{-04}$	1.52 $\times \;10^{-04}$	7.76 $\times \;10^{-05}$	1.48 $\times \;10^{-04}$	1.10 $\times \;10^{-04}$	2.77 $\times \;10^{-05}$
F6	mean	−1.26 $\times \;10^{+04}$	−1.25 $\times \;10^{+04}$	−1.25 $\times \;10^{+04}$	−1.25 $\times \;10^{+04}$	−1.26 $\times \;10^{+04}$	−1.26 $\times \;10^{+04}$
	best	−1.26 $\times \;10^{+04}$	−1.26 $\times \;10^{+04}$	−1.26 $\times \;10^{+04}$	−1.26 $\times \;10^{+04}$	−1.26 $\times \;10^{+04}$	−1.26 $\times \;10^{+04}$
	worst	−1.26 $\times \;10^{+04}$	−1.23 $\times \;10^{+04}$	−1.16 $\times \;10^{+04}$	−1.16 $\times \;10^{+04}$	− 1.26 $\times \;10^{+04}$	−1.26 $\times \;10^{+04}$
	std	1.43 $\times \;10^{+00}$	8.41 $\times \;10^{+01}$	2.25 $\times \;10^{+02}$	1.75 $\times \;10^{+02}$	1.07 $\times \;10^{+00}$	2.89 $\times \;10^{-01}$
F7	mean	0	0	0	0	0	0
	best	0	0	0	0	0	0
	worst	0	0	0	0	0	0
	std	0	0	0	0	0	0
F8	mean	8.88 $\times \;10^{-16}$	8.88 $\times \;10^{-16}$	8.88 $\times \;10^{-16}$	8.88 $\times \;10^{-16}$	8.88 $\times \;10^{-16}$	8.88 $\times \;10^{-16}$
	best	8.88 $\times \;10^{-16}$	8.88 $\times \;10^{-16}$	8.88 $\times \;10^{-16}$	8.88 $\times \;10^{-16}$	8.88 $\times \;10^{-16}$	8.88 $\times \;10^{-16}$
	worst	8.88 $\times \;10^{-16}$	8.88 $\times \;10^{-16}$	8.88 $\times \;10^{-16}$	8.88 $\times \;10^{-16}$	8.88 $\times \;10^{-16}$	8.88 $\times \;10^{-16}$
	std	0	0	0	0	0	0
F9	mean	1.43 $\times \;10^{+00}$	1.29 $\times \;10^{+00}$	1.86 $\times \;10^{+00}$	2.08 $\times \;10^{+00}$	1.30 $\times \;10^{+00}$	9.98 $\times \;10^{-01}$
	best	9.98 $\times \;10^{-01}$	9.98 $\times \;10^{-01}$	9.98 $\times \;10^{-01}$	9.98 $\times \;10^{-01}$	9.98 $\times \;10^{-01}$	9.98 $\times \;10^{-01}$
	worst	5.93 $\times \;10^{+00}$	5.93 $\times \;10^{+00}$	2.98 $\times \;10^{+00}$	5.93 $\times \;10^{+00}$	2.98 $\times \;10^{+00}$	1.01 $\times \;10^{+00}$
	std	1.26 $\times \;10^{+00}$	9.40 $\times \;10^{-01}$	9.30 $\times \;10^{-01}$	1.80 $\times \;10^{+00}$	5.30 $\times \;10^{-01}$	1.57 $\times \;10^{-03}$
F10	mean	3.89 $\times \;10^{-04}$	7.04 $\times \;10^{-04}$	5.32 $\times \;10^{-04}$	3.41 $\times \;10^{-04}$	3.45 $\times \;10^{-04}$	3.40 $\times \;10^{-04}$
	best	3.09 $\times \;10^{-04}$	3.16 $\times \;10^{-04}$	3.37 $\times \;10^{-04}$	3.08 $\times \;10^{-04}$	3.10 $\times \;10^{-04}$	3.02 $\times \;10^{-04}$
	worst	1.51 $\times \;10^{-03}$	1.79 $\times \;10^{-03}$	1.33 $\times \;10^{-03}$	3.99 $\times \;10^{-04}$	4.51 $\times \;10^{-04}$	4.00 $\times \;10^{-04}$
	std	2.15 $\times \;10^{-04}$	5.30 $\times \;10^{-04}$	2.03 $\times \;10^{-04}$	2.96 $\times \;10^{-05}$	3.82 $\times \;10^{-05}$	1.78 $\times \;10^{-05}$
F11	mean	−5.03 $\times \;10^{+00}$	−5.12 $\times \;10^{+00}$	−6.33 $\times \;10^{+00}$	−6.61 $\times \;10^{+00}$	−5.47 $\times \;10^{+00}$	−1.03 $\times \;10^{+01}$
	best	−5.13 $\times \;10^{+00}$	−5.11 $\times \;10^{+00}$	−1.03 $\times \;10^{+01}$	−1.05 $\times \;10^{+01}$	−1.03 $\times \;10^{+01}$	−1.05 $\times \;10^{+01}$
	worst	−2.41 $\times \;10^{+00}$	−5.13 $\times \;10^{+00}$	−5.06 $\times \;10^{+00}$	−5.00 $\times \;10^{+00}$	−5.12 $\times \;10^{+00}$	−9.50 $\times \;10^{+00}$
	std	4.94 $\times \;10^{-01}$	4.49 $\times \;10^{-03}$	1.79 $\times \;10^{+00}$	2.38 $\times \;10^{+00}$	1.30 $\times \;10^{+00}$	2.25 $\times \;10^{-01}$

**Table 5 biomimetics-08-00428-t005:** The calculated optimization value of each test function with different algorithms.

Function	Index	GWO	WOA	PWOA	HHO	MHHO	ADHHO	IHHO
F1	mean	3.59 $\times \;10^{-27}$	1.60 $\times \;10^{-73}$	8.71 $\times \;10^{-92}$	1.50 $\times \;10^{-95}$	1.88 $\times \;10^{-120}$	5.93 $\times \;10^{-194}$	3.25 $\times \;10^{-221}$
	best	1.97 $\times \;10^{-29}$	2.58 $\times \;10^{-87}$	1.48 $\times \;10^{-111}$	1.28 $\times \;10^{-109}$	1.44 $\times \;10^{-141}$	3.57 $\times \;10^{-221}$	1.62 $\times \;10^{-238}$
	worst	9.37 $\times \;10^{-26}$	3.04 $\times \;10^{-72}$	2.61 $\times \;10^{-90}$	3.63 $\times \;10^{-94}$	5.64 $\times \;10^{-119}$	3.35 $\times \;10^{-193}$	4.20 $\times \;10^{-220}$
	std	1.70 $\times \;10^{-26}$	6.20 $\times \;10^{-73}$	4.77 $\times \;10^{-91}$	6.75 $\times \;10^{-95}$	1.03 $\times \;10^{-119}$	2.14 $\times \;10^{-193}$	0.99 $\times \;10^{-220}$
F2	mean	9.43 $\times \;10^{-17}$	3.71 $\times \;10^{-50}$	2.72 $\times \;10^{-71}$	3.95 $\times \;10^{-51}$	1.55 $\times \;10^{-62}$	5.70 $\times \;10^{-100}$	2.16 $\times \;10^{-108}$
	best	2.25 $\times \;10^{-17}$	4.72 $\times \;10^{-57}$	2.67 $\times \;10^{-81}$	5.45 $\times \;10^{-59}$	3.13 $\times \;10^{-72}$	1.81 $\times \;10^{-111}$	9.26 $\times \;10^{-124}$
	worst	2.80 $\times \;10^{-16}$	1.01 $\times \;10^{-48}$	5.91 $\times \;10^{-70}$	8.91 $\times \;10^{-50}$	4.32 $\times \;10^{-61}$	4.08 $\times \;10^{-99}$	6.48 $\times \;10^{-107}$
	std	6.09 $\times \;10^{-17}$	1.84 $\times \;10^{-49}$	1.08 $\times \;10^{-70}$	1.64 $\times \;10^{-50}$	7.86 $\times \;10^{-62}$	9.43 $\times \;10^{-100}$	1.18 $\times \;10^{-107}$
F3	mean	1.72 $\times \;10^{-05}$	4.95 $\times \;10^{+04}$	5.25 $\times \;10^{+04}$	5.54 $\times \;10^{-71}$	1.72 $\times \;10^{-98}$	8.91 $\times \;10^{-129}$	3.85 $\times \;10^{-172}$
	best	9.62 $\times \;10^{-10}$	1.35 $\times \;10^{+04}$	2.73 $\times \;10^{+04}$	5.14 $\times \;10^{-103}$	1.34 $\times \;10^{-125}$	4.21 $\times \;10^{-166}$	3.39 $\times \;10^{-196}$
	worst	2.11 $\times \;10^{-04}$	7.34 $\times \;10^{+04}$	9.07 $\times \;10^{+04}$	1.65 $\times \;10^{-69}$	4.10 $\times \;10^{-97}$	1.21 $\times \;10^{-128}$	1.15 $\times \;10^{-170}$
	std	4.21 $\times \;10^{-05}$	1.30 $\times \;10^{+04}$	1.63 $\times \;10^{+04}$	3.02 $\times \;10^{-70}$	7.66 $\times \;10^{-98}$	3.01 $\times \;10^{-130}$	2.10 $\times \;10^{-171}$
F4	mean	7.52 $\times \;10^{-07}$	5.36 $\times \;10^{+01}$	5.64 $\times \;10^{+01}$	3.16 $\times \;10^{-48}$	7.44 $\times \;10^{-63}$	9.85 $\times \;10^{-87}$	6.19 $\times \;10^{-112}$
	best	6.17 $\times \;10^{-08}$	2.30 $\times \;10^{-01}$	3.05 $\times \;10^{+00}$	1.21 $\times \;10^{-56}$	1.17 $\times \;10^{-71}$	5.74 $\times \;10^{+112}$	5.35 $\times \;10^{-121}$
	worst	2.70 $\times \;10^{-06}$	8.78 $\times \;10^{+01}$	9.14 $\times \;10^{+01}$	8.18 $\times \;10^{-47}$	1.79 $\times \;10^{-61}$	1.61 $\times \;10^{-89}$	1.33 $\times \;10^{-110}$
	std	6.33 $\times \;10^{-07}$	2.64 $\times \;10^{+01}$	2.73 $\times \;10^{+01}$	1.49 $\times \;10^{-47}$	3.33 $\times \;10^{-62}$	1.68 $\times \;10^{-90}$	2.44 $\times \;10^{-111}$
F5	mean	1.96 $\times \;10^{-03}$	5.06 $\times \;10^{-03}$	2.30 $\times \;10^{-03}$	1.58 $\times \;10^{-04}$	1.42 $\times \;10^{-04}$	1.01 $\times \;10^{-04}$	3.74 $\times \;10^{-05}$
	best	4.54 $\times \;10^{-04}$	1.86 $\times \;10^{-04}$	1.04 $\times \;10^{-04}$	9.70 $\times \;10^{-06}$	7.34 $\times \;10^{-06}$	2.79 $\times \;10^{-06}$	2.33 $\times \;10^{-06}$
	worst	4.20 $\times \;10^{-03}$	1.69 $\times \;10^{-02}$	8.45 $\times \;10^{-03}$	5.82 $\times \;10^{-04}$	6.07 $\times \;10^{-04}$	2.63 $\times \;10^{-04}$	9.57 $\times \;10^{-05}$
	std	9.39 $\times \;10^{-04}$	4.65 $\times \;10^{-03}$	2.24 $\times \;10^{-03}$	1.75 $\times \;10^{-04}$	1.46 $\times \;10^{-04}$	9.93 $\times \;10^{-05}$	2.77 $\times \;10^{-05}$
F6	mean	−6.08 $\times \;10^{+03}$	−1.00 $\times \;10^{+04}$	−9.75 $\times \;10^{+03}$	−1.26 $\times \;10^{+04}$	−1.26 $\times \;10^{+04}$	−1.26 $\times \;10^{+04}$	−1.26 $\times \;10^{+04}$
	best	−7.31 $\times \;10^{+03}$	−1.26 $\times \;10^{+04}$	−1.26 $\times \;10^{+04}$	−1.26 $\times \;10^{+04}$	−1.26 $\times \;10^{+04}$	−1.26 $\times \;10^{+04}$	−1.26 $\times \;10^{+04}$
	worst	−3.19 $\times \;10^{+03}$	−7.44 $\times \;10^{+03}$	−6.98 $\times \;10^{+03}$	−1.26 $\times \;10^{+04}$	−1.26 $\times \;10^{+04}$	−1.26 $\times \;10^{+04}$	−1.26 $\times \;10^{+04}$
	std	9.71 $\times \;10^{+02}$	1.79 $\times \;10^{+03}$	1.77 $\times \;10^{+03}$	1.43 $\times \;10^{+00}$	5.77 $\times \;10^{-01}$	3.91 $\times \;10^{-01}$	2.89 $\times \;10^{-01}$
F7	mean	3.18 $\times \;10^{+00}$	0	0	0	0	0	0
	best	5.68 $\times \;10^{-14}$	0	0	0	0	0	0
	worst	1.02 $\times \;10^{+01}$	0	0	0	0	0	0
	std	2.74 $\times \;10^{+00}$	0	0	0	0	0	0
F8	mean	1.04 $\times \;10^{-13}$	4.68 $\times \;10^{-15}$	4.20 $\times \;10^{-15}$	8.88 $\times \;10^{-16}$	8.88 $\times \;10^{-16}$	8.88 $\times \;10^{-16}$	8.88 $\times \;10^{-16}$
	best	7.55 $\times \;10^{-14}$	8.88 $\times \;10^{-16}$	8.88 $\times \;10^{-16}$	8.88 $\times \;10^{-16}$	8.88 $\times \;10^{-16}$	8.88 $\times \;10^{-16}$	8.88 $\times \;10^{-16}$
	worst	7.55 $\times \;10^{-14}$	7.99 $\times \;10^{-15}$	7.99 $\times \;10^{-15}$	8.88 $\times \;10^{-16}$	8.88 $\times \;10^{-16}$	8.88 $\times \;10^{-16}$	8.88 $\times \;10^{-16}$
	std	1.69 $\times \;10^{-14}$	2.46 $\times \;10^{-15}$	2.63 $\times \;10^{-15}$	0	0	0	0
F9	mean	5.17 $\times \;10^{+00}$	3.90 $\times \;10^{+00}$	6.34 $\times \;10^{+00}$	1.43 $\times \;10^{+00}$	1.33 $\times \;10^{+00}$	1.32 $\times \;10^{+00}$	9.98 $\times \;10^{-01}$
	best	9.98 $\times \;10^{-01}$	9.98 $\times \;10^{-01}$	9.98 $\times \;10^{-01}$	9.98 $\times \;10^{-01}$	9.98 $\times \;10^{-01}$	9.98 $\times \;10^{-01}$	9.98 $\times \;10^{-01}$
	worst	1.27 $\times \;10^{+01}$	1.08 $\times \;10^{+01}$	1.27 $\times \;10^{+01}$	5.93 $\times \;10^{+00}$	5.93 $\times \;10^{+00}$	1.92 $\times \;10^{+00}$	1.01 $\times \;10^{+00}$
	std	4.54 $\times \;10^{+00}$	3.94 $\times \;10^{+00}$	4.74 $\times \;10^{+00}$	1.26 $\times \;10^{+00}$	9.47 $\times \;10^{-01}$	9.41 $\times \;10^{-01}$	1.57 $\times \;10^{-03}$
F10	mean	3.73 $\times \;10^{-03}$	8.83 $\times \;10^{-04}$	7.04 $\times \;10^{-04}$	3.89 $\times \;10^{-04}$	4.04 $\times \;10^{-04}$	3.52 $\times \;10^{-04}$	3.40 $\times \;10^{-04}$
	best	3.07 $\times \;10^{-04}$	3.08 $\times \;10^{-04}$	3.16 $\times \;10^{-04}$	3.09 $\times \;10^{-04}$	3.08 $\times \;10^{-04}$	3.07 $\times \;10^{-04}$	3.02 $\times \;10^{-04}$
	worst	2.04 $\times \;10^{-02}$	2.25 $\times \;10^{-03}$	2.25 $\times \;10^{-03}$	1.51 $\times \;10^{-03}$	1.54 $\times \;10^{-03}$	2.01 $\times \;10^{-03}$	4.00 $\times \;10^{-04}$
	std	7.57 $\times \;10^{-03}$	5.25 $\times \;10^{-04}$	4.18 $\times \;10^{-04}$	2.15 $\times \;10^{-04}$	2.75 $\times \;10^{-04}$	0.24 $\times \;10^{-04}$	1.78 $\times \;10^{-05}$
F11	mean	−1.04 $\times \;10^{+01}$	−6.97 $\times \;10^{+00}$	−5.87 $\times \;10^{+00}$	−5.03 $\times \;10^{+00}$	−5.26 $\times \;10^{+00}$	−6.35 $\times \;10^{+00}$	−1.03 $\times \;10^{+01}$
	best	−1.05 $\times \;10^{+01}$	−1.05 $\times \;10^{+01}$	−1.05 $\times \;10^{+01}$	−5.13 $\times \;10^{+00}$	−1.04 $\times \;10^{+01}$	−1.04 $\times \;10^{+01}$	−1.05 $\times \;10^{+01}$
	worst	−5.17 $\times \;10^{+00}$	−2.42 $\times \;10^{+00}$	−1.67 $\times \;10^{+00}$	−2.41 $\times \;10^{+00}$	−3.77 $\times \;10^{+00}$	−4.98 $\times \;10^{+00}$	−9.50 $\times \;10^{+00}$
	std	9.79 $\times \;10^{-01}$	3.03 $\times \;10^{+00}$	2.70 $\times \;10^{+00}$	4.94 $\times \;10^{-01}$	1.01 $\times \;10^{+00}$	2.17 $\times \;10^{+00}$	2.25 $\times \;10^{-01}$

**Table 6 biomimetics-08-00428-t006:** Set path planning parameters.

Size	Parameter	Value
20×20	Number of population	20
	The number of iterations of the path	50
	Number of iterations of the population	500
60×60	Number of population	60
	The number of iterations of the path	50
	Number of iterations of the population	1000

**Table 7 biomimetics-08-00428-t007:** Path length comparison.

Size	Index	A-Star	EGWO	HIWOA	HHO	MHHO	ADHHO	IHHO
20 × 20	Mean	/	29.00	31.20	28.86	34.09	30.25	28.00
	Best	33.05	28.10	29.25	28.13	28.75	27.74	27.30
	Worst	/	31.06	34.18	30.14	44.91	35.12	29.32
60 × 60	Mean	/	91.97	96.92	90.50	94.37	93.11	85.39
	Best	/	89.39	92.71	88.16	87.82	88.73	84.47
	Worst	/	95.93	104.79	93.42	100.51	97.08	87.62

## Data Availability

Not applicable.

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
