# Peer review of "An Improved Harris Hawks Optimization Algorithm and Its Application in Grid Map Path Planning"

_biomimetics, 2023, doi:10.3390/biomimetics8050428_

Round 1

Reviewer 1 Report

Since the subject of study is a rapidly expanding research area on the agenda, the importance of the proposed method increases even more. The work is a good job but definitely needs some revisions.

1- Explain clearly what the problem is.

2- The contributions of the study should be presented clearly.

3- Why is path planning done on the grid map? What is the performance and success of your proposed method in blind environments?

4- Is the path planning 2D or 3D? needs to be stated clearly.

5- What is your motivation for using the Harris Hawks Optimization (HHO) algorithm? For example, what is the advantage over algorithms such as GWO, SCSO, SCA, and so on?

6- Indicate why you have chosen the nature-inspired approach by focusing on the existing categories in the literature for solving the path planning problem. Consider helpful case studies such as Adapted-RRT method. Also, it is important and helpful for you to refer to the following study:

https://jit.ndhu.edu.tw/article/view/2539

7- Analyze your results with diversity and exploration/exploitation parameters. The following study will help you, you can cite it.

https://doi.org/10.3390/math11102340

8- Why are parameters such as speed and obstacle not taken into account in your mathematical model?

9- Due to the proposed method's working mechanism being somewhat similar to AGTO and WOA algorithms, include these algorithms in your comparisons. Also, at least one chaos-based method can be included in the comparisons. Note: Since the WOA study is from 2015, it is considered old and you can add a new and hybrid variant of it published in good journals. For example, the Reinforcement Learning-WOA (RLWOA) algorithm. 

10- Add the limitations of your method. Also, include the future works to last section.

minor

Reviewer 2 Report

The comments are in the attached file.

The English correction is also in the attached file

Reviewer 3 Report

1- This is interesting research, and the topic is within the scope of the journal; the authors are encouraged to extend the introduction by using studies in this area and the application of different optimization algorithms in other areas of engineering as well to justify their research method, the current introduction is insufficient. The following studies are recommended:

https://doi.org/10.2478/jaes-2018-0010

https://doi.org/10.12989/sss.2020.26.5.559

 2- Authors should express clearly their main problem and their solution, information must be included in comprehensive detail.

3- Authors must explain why they have chosen this optimization method and what are the benefits of using this method comparing with others.

4-Authors also must express the drawbacks of this method compared with others and express its advantages, this information must be in the manuscript

5-  in section 2.2 ( mathematical model ) more elaboration is required 

6- section 4,1 and 4.2 should be expanded to justify the method 

7- Figs 6 and 7 should be changed to higher quality and larger images

8- references are insufficient

authors are encouraged to review the paper and correct all typos

Round 2

Reviewer 1 Report

The article has been well revised overall, but some comments need to be answered more clearly. Therefore, this study can be accepted provided that the following new comments are answered/revised.

Responses and corrections to 10 comments made in the first round:

A) Comments for 1 to 4 are well answered.

B) Question 5 is also well answered, but no reference is made to the relevant optimization studies presented. These should be added. For at least three examples given. 

C) The 6th comment was not well answered and the references were not taken into account/reflected. This question should be reexamined.

D) Comments 7, 8, and 10 have been carefully and meticulously answered and added to the article.

E) I could neither see the results of the 9th comment in Table 5 nor its reference. You used RLWOA but not fully reflected. If it should be RLWOA, it was written as PWOA mistakenly and its reference is also missing. Reference of the related work is: https://doi.org/10.1016/j.knosys.2021.107044

Reviewer 2 Report

The author adequately replied my comments academically. However, one serious problem is the English writing. It seems the author rushed to make the changes. These changes contain a large amount of careless grammar and writing errors. Many of the sentences are not comprehensible. The attached file has my edits, but they are by no means of all the edits needed. To the very least, the first sentence of the abstract needs to clear and grammarly correct before publication.

The English writing of the newly added content needs significantly improvement. Some of the edits are attached in my referee letter.

Reviewer 3 Report

Accept

Author Response

Thanks for the positive comments.